# EVALUATING THE EVALUATORS: ARE CURRENT FEW-SHOT LEARNING BENCHMARKS FIT FOR PURPOSE?

## ABSTRACT

Numerous benchmarks for Few-Shot Learning have been proposed in the last decade. However all of these benchmarks focus on performance averaged over many tasks, and the question of how to reliably evaluate and tune models trained for individual few-shot tasks has not been addressed. This paper presents the first investigation into task-level validation—a fundamental step when deploying a model. We measure the accuracy of performance estimators in the few-shot setting, consider strategies for model selection, and examine the reasons for the failure of evaluators usually thought of as being robust. We conclude that cross-validation with a low number of folds is the best choice for directly estimating the performance of a model, whereas using bootstrapping or cross validation with a large number of folds is better for model selection purposes. Overall, we find that with current methods, benchmarks, and validation strategies, one can not get a reliable picture of how effectively methods perform on individual tasks.

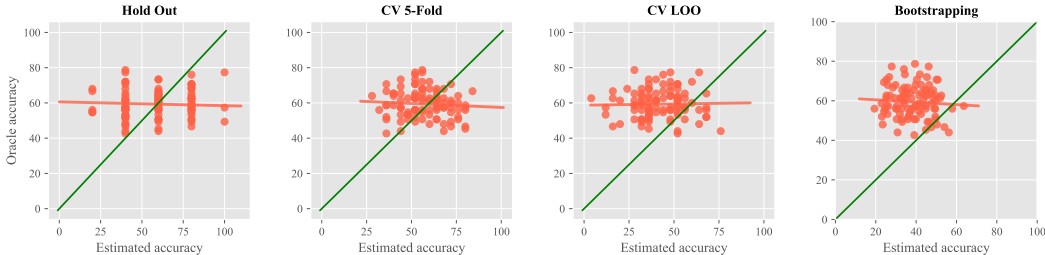

Figure 1: Scatter plots of model accuracy on query set versus accuracy estimated using only the support set. The ideal estimator would have the points almost co-linear and lying approximately on the diagonal line (green). All estimators have very high bias (points are off diagonal) and variance (no co-linearity), indicating they provide poor estimates of performance.

## 1 INTRODUCTION

Deep learning excels on many tasks where large-scale datasets are available for training (LeCun et al., 2015). However, many standard deep learning techniques struggle to construct high accuracy models when only a very small number of training examples are available. This is a serious impediment to the broader uptake of machine learning in domains where web-scale data are not available. In many domains, such as medicine and security, it is common to suffer from data scarcity issues due to a multitude of resource constraints and the rarity of the events being modelled. The Few Shot-Learning (FSL) paradigm, which focuses on enabling models to generalise well with little data through the use of transferred prior knowledge, has gained relevance in an attempt to overcome these challenges. A significant amount of attention has been given to FSL and related meta-learning research in the last decade (Wang et al., 2020; Hospedales et al., 2021), with a large number of methods and benchmarks proposed in application domains ranging from visual recognition systems for robots to identifying therapeutic properties of molecules (Xie et al., 2018; Stanley et al., 2021).

Even though many learning algorithms have been developed in this area, and great efforts have been directed towards improving model performance in FSL scenarios (Finn et al., 2017; Snell

et al., 2017; Hospedales et al., 2021), the best practices for how to evaluate models and design benchmarks for this paradigm remain relatively unexplored. In typical academic benchmark setups, performance estimation often relies on the existence of test ("query") sets that are several times larger than training ("support") sets—a desideratum that is clearly not satisfied in realistic FSL scenarios. Moreover, performance is averaged over a large number of learning problem, leading to accuracy estimates that cannot be used as a validation score for a single problem ("episode") of interest. While these experimental design decisions make sense when assessing the general efficacy of a learning algorithm, they are not helpful when considering how one might validate or select a model for a real FSL problem with a single few-shot support set. Even in the case where a good estimate of model performance is not required, little attention has been given to robust methods for selecting the best model from some prospective pool of models when little data is available. The shortage of model selection procedures for FSL settings means that one cannot reliably perform hyperparameter optimisation or select which FSL algorithm to use.

We hypothesise that standard model evaluation procedures are not effective in the few-shot learning regime—much like how standard learning algorithms are not well-suited to few-shot learning. We experimentally investigate this hypothesis by analysing the behaviour of commonly used model evaluation processes. In addition to studying the accuracy of the performance estimates, we also consider how well these potentially noisy estimates can be slotted into model selection pipelines where one cares only about the relative performance of different models. We analyse failure modes of these methods and show that, should evaluation methods be improved, one could see both a risk reduction for FSL deployments and also better models, due to the ability to do model selection reliably.

We answer several concrete questions about the state of methods currently available for evaluating models trained in a FSL setting:

**Q1** *How accurately can we estimate the performance of task-level models trained in the FSL regime?* There are no combinations of learning algorithms and evaluators that are able to produce reliable performance estimates, but we find that 5-fold cross-validation is the best of the bad options.

**Q2** *Are rankings produced by existing evaluators well-correlated with the true performance rankings of models?* Current model evaluation procedures do not provide reliable rankings at the per-episode level, but methods based on resampling with a large number of iterations are most reliable.

**Q3** *By how much could performance in FSL be improved by incorporating accurate model selection procedures?* Our results show that there is still a lot of room for improvement in the case of model selection, as evidenced by the large gap between performance obtained via current model selection methods and performance from using an oracle evaluator to perform model selection.

## 2 RELATED WORK

Current practices for evaluating FSL arose from the work on developing new learning algorithms (Lake et al., 2015; Ravi & Larochelle, 2017; Ren et al.; Triantafillou et al., 2020; Ullah et al., 2022), and the aim has consistently been to determine the general efficacy of new learning algorithms. The experimental setup used by these works employ a large number of downstream FSL tasks (drawn from a so-called meta-test set), each of which has a test set several times larger than the associated training set. The endpoint measured and compared when making conclusions is the average accuracy across the episodes in the meta-test set. While such an experimental setup is sensible when assessing the "average-case" performance of a FSL method, in this work we address the more classic model validation and selection problem encountered when deploying machine learning models, which is currently neglected in existing FSL benchmarks. I.e., *Given a single specific few-shot learning task defined by a small training/support set, how can we estimate the performance on the unseen test/query set?* This is crucial in order to perform model selection, and to validate whether predictions of the few-shot learner are safe to act upon or not. Recent work from application domains that require few-shot learning indicate that the lack of reliable evaluation procedures is a major blocker for deployment (Varoquaux & Cheplygina, 2022).

Some prior work on studying the failure modes of FSL has identified that the variance of accuracy across episodes is typically very high (Dhillon et al., 2020; Agarwal et al., 2021; Basu et al., 2023). Work in this area has focused on identifying "hard" episodes by constructing support sets that lead

to pathological performance on a given query set. The focus of this existing work has been on discovering trends in the types of downstream tasks where FSL methods do not work, whereas our motivation is to explore: i) what processes can be followed to determine whether a particular model will perform as required; and ii) how can we reliably select the best model from some set of potential models. These specific questions are currently unaddressed by existing literature.

## 3   FEW-SHOT MODEL VALIDATION AND SELECTION

Modern few-shot learning research considers a two-level data generating process. The top level corresponds to a distribution over FSL tasks, and the bottom level represents the data generation process for an individual task. More precisely, consider a top level *environment* distribution, $\mathcal{E}$, from which one can sample a distributions, $\mathcal{D}_i$, associated with a task indexed by $i$. One can then sample data from $\mathcal{D}_i$ to generate examples for task $i$. We first outline the existing assumptions about data availability and evaluation procedures used in the literature, which we refer to as Aggregated Evaluation (AE). We then discuss how this can be augmented to better match real-world FSL scenarios, and then outline the Task-Level Evaluation (TLE) and Task-Level Model Selection (TLMS) protocols that we experimentally investigate.

**Aggregated Evaluation (AE)**   In conventional Aggregated Evaluation, one samples a collection of distributions, $\mathbb{D} = \{\mathcal{D}_i \sim \mathcal{E}\}_{i=1}^t$, and then for each of these distributions one constructs a meta-test episode consisting of a support set, $S_i = \{(\boldsymbol{x}_j, y_j) \sim \mathcal{D}_i\}_{j=1}^n$, and a query set, $Q_i = \{(\boldsymbol{x}_j, y_j) \sim \mathcal{D}_i\}_{j=1}^m$. It is typical for the size of the query set, $m$, to be several times larger than the size of the support set, $n$. We observe that in a true FSL setting this is an unrealistic assumption.

Current standard practice when evaluating FSL methods is to sample $t$ episodes. For each episode, a model is trained using the support set and then a performance estimate is computed using the query set, $\hat{\mu}_{\mathcal{D}_i} = \frac{1}{m} \sum_{\boldsymbol{x}_j, y_j \in Q_i} \mathbf{1}(y_j = h_{S_i}(\boldsymbol{x}_j))$, where $h_{S_i}$ denotes a model trained on $S_i$ and $\mathbf{1}(\cdot)$ is the indicator function. Finally, the performance of the learning algorithm is summarised by aggregating over all the meta-test episodes, $\hat{\mu}_{\mathcal{E}} = \frac{1}{t} \sum_{i=1}^t \hat{\mu}_{\mathcal{D}_i}$.

Using $\hat{\mu}_{\mathcal{E}}$ to evaluate the performance of a FSL algorithm makes sense if the downstream application involves a large number of different FSL problems, and the success of the overall system is dependent on being accurate on average. Examples of such applications include recommender systems and personalised content tagging, where each episode corresponds to a single user session. The success of such personalisation systems depends on being accurate for most user sessions, but a poor experience for a small number of sessions (i.e., episodes) is acceptable. In contrast, applications of FSL in medicine or security are not well-suited to the aggregated evaluation provided by $\hat{\mu}_{\mathcal{E}}$. In these settings, each episode might correspond to recognising the presence of a specific pathology or security threat, and poor performance on such episodes would translate to systematic misdiagnosis and critical security vulnerabilities. Such outcomes would be considered a catastrophic system failure.

**Task-Level Evaluation (TLE)**   We investigate Task-Level Evaluation, which serves a purpose distinct from AE. While AE is typically undertaken to assess the general efficacy of a FSL algorithm, the purpose of TLE is to determine the performance of a model trained for a particular episode. In many real-world applications of FSL, one must have accurate estimates of the performance of models trained for each episode. Moreover, in realistic situations there is no labelled query set for evaluating a model, so both the model fitting and model evaluation must be done with the support set. Just as FSL requires specialised learning algorithms, we argue that TLE of FSL requires specialised performance estimators.

We investigate three common approaches to evaluating machine learning models, which we summarise below.

*Hold-out (Witten et al., 2016)* The Hold-out evaluation method, which is commonly used in the deep learning literature, requires a portion of the data to be set aside so it can be used to test the model. In particular, one sets aside $n$ samples from the support set, trains a model on the $m - n$ remaining samples, and uses the $n$ held-out data points to estimate performance.

Table 1: 5-way/5-shot few-shot learning accuracy considering a 95% confidence interval: Actual (Oracle) vs as predicted by various estimators. Most methods under-estimate accuracy compared to Oracle.

| | Model | Oracle | Hold-Out | CV 5 Fold | CV LOO | Bootstrapping |
|---|---|---|---|---|---|---|
| CIFAR-FS | **Baseline** | $71.17 \pm 0.727$ | $64.50 \pm 2.235$ | $69.28 \pm 1.040$ | $62.47 \pm 1.122$ | $54.09 \pm 0.895$ |
| | **Baseline++** | $71.38 \pm 0.755$ | $66.28 \pm 2.262$ | $70.41 \pm 1.017$ | $63.80 \pm 1.145$ | $56.91 \pm 0.969$ |
| | **ProtoNet** | $71.65 \pm 0.755$ | $70.30 \pm 1.717$ | $70.09 \pm 1.006$ | $65.43 \pm 1.089$ | $57.31 \pm 0.941$ |
| | **MAML** | $69.51 \pm 0.762$ | $69.93 \pm 1.819$ | $71.41 \pm 1.119$ | $16.79 \pm 0.667$ | $42.39 \pm 0.666$ |
| | **R2D2** | $72.06 \pm 0.751$ | $71.37 \pm 1.609$ | $70.22 \pm 1.021$ | $62.24 \pm 1.172$ | $55.12 \pm 0.905$ |
| miniImageNet | **Baseline** | $59.36 \pm 0.646$ | $57.17 \pm 1.765$ | $57.15 \pm 1.040$ | $44.80 \pm 1.094$ | $38.34 \pm 0.723$ |
| | **Baseline++** | $64.07 \pm 0.632$ | $63.30 \pm 1.683$ | $63.31 \pm 0.978$ | $21.28 \pm 0.957$ | $23.97 \pm 0.452$ |
| | **ProtoNet** | $66.12 \pm 0.676$ | $63.73 \pm 1.699$ | $63.98 \pm 1.000$ | $51.69 \pm 1.177$ | $37.67 \pm 0.740$ |
| | **MAML** | $59.76 \pm 0.717$ | $48.11 \pm 2.286$ | $59.81 \pm 1.135$ | $17.39 \pm 0.684$ | $16.28 \pm 0.350$ |
| | **R2D2** | $63.58 \pm 0.658$ | $62.67 \pm 1.682$ | $62.33 \pm 1.001$ | $48.58 \pm 1.126$ | $40.54 \pm 0.734$ |
| MetaAlbum | **Baseline** | $59.36 \pm 1.688$ | $54.39 \pm 2.658$ | $57.43 \pm 1.792$ | $51.58 \pm 1.919$ | $45.68 \pm 1.695$ |
| | **Baseline++** | $57.99 \pm 1.724$ | $53.11 \pm 2.616$ | $56.78 \pm 1.837$ | $48.84 \pm 2.005$ | $43.85 \pm 1.701$ |
| | **ProtoNet** | $47.74 \pm 1.689$ | $46.73 \pm 2.233$ | $46.30 \pm 1.813$ | $44.07 \pm 1.849$ | $39.44 \pm 1.665$ |
| | **MAML** | $47.10 \pm 1.736$ | $42.78 \pm 2.603$ | $47.50 \pm 1.921$ | $15.24 \pm 0.819$ | $28.68 \pm 1.088$ |
| | **R2D2** | $52.91 \pm 1.670$ | $51.30 \pm 2.242$ | $51.39 \pm 1.779$ | $44.70 \pm 1.890$ | $39.39 \pm 1.599$ |

*Cross-validation (CV) (Stone, 1974)* The cross-validation method splits the support set in $k$ folds and repeats model training $k$ time. For each repetition, a different fold is used for testing the performance of the model, and the other $k-1$ folds are used to train the model. The mean performance across all iterations is used as a performance estimate for a model trained on all the data. When $k$ is set to the number of data points this method is referred to as cross-validation leave-one-out (CV-LOO).

*Bootstrapping (Efron, 1979)* Bootstrap sampling is a technique which consists of randomly drawing sample data with replacement from a given set, resulting in a new sample with the same size as the original set, and an out-of-bag set composed of the data points that where not chosen during the sampling process. This method is used in our bootstrapping evaluator to create a new support sample from the support set and a new query set composed by the out-of-bag data points. The process is repeated and the result of each repetition averaged to give a final performance estimate.

Estimators are typically evaluated in terms of bias and variance, defined respectively as

$$\text{Bias}(\hat{\mu}) = \mathbb{E}[\hat{\mu}] - \mathbb{E}[\mu], \qquad \text{Var}(\hat{\mu}) = \mathbb{E}\left[(\hat{\mu} - \mathbb{E}[\hat{\mu}])^2\right],$$

where $\hat{\mu}$ is a performance estimate and $\mu$ is the true performance. To provide a more directly interpretable metric than variance for measuring the expected deviation from the true performance of a model, we consider the Mean Absolute Error (MAE) between a performance estimator and the true quantity,

$$\text{MAE}(\hat{\mu}) = \mathbb{E}\left[|\hat{\mu} - \mathbb{E}[\mu]|\right].$$

In practice, the Bias and MAE are estimated using episodes from the meta-test set.

**Task-Level Model Selection** One of the common use-cases for performance estimators is in the model selection process, where they are used to rank the relative performance of a set of candidate models, either for deployment or as part of the hyperparameter optimisation process. We observe that accurate estimates of the performance of models may not be needed for model selection, as long as similar types of errors as made for all the prospective models. For example, if a performance estimator is systematically biased, but does not exhibit much variance, then the resulting rankings should be reliable, even though the estimate of the performance is not.

## 4 EXPERIMENTS

We conduct experiments to determine which evaluation techniques are most reliable for: i) estimating the performance of a model; and ii) selecting the best performing model from some pool of prospective models. We consider both the standard in-domain setting, and the more realistic cross-domain setting. Following these experiments, we investigate the underlying reasons why evaluators traditionally thought of as being quite robust can fail in the FSL setting.

We conduct experiments on datasets and learners popular in both the meta-learning (Hospedales et al., 2021) and vision & language transfer learning (Radford et al., 2021) family of approaches to few-shot learning. Specifically, we use:

Table 2: Per episode accuracy MAE between accuracy of oracle and accuracy of each estimator (5-way 5-shot setting)

| | Model | Hold-Out | CV 5 Fold | CV LOO | Bootstrapping |
|---|---|---|---|---|---|
| CIFAR-FS | **Baseline** | 24.06 | 12.38 | 14.41 | 18.65 |
| | **Baseline++** | 23.75 | 12.49 | 14.59 | 17.27 |
| | **ProtoNet** | 16.75 | 8.67 | 10.16 | 15.00 |
| | **MAML** | 19.74 | 14.00 | 52.72 | 27.25 |
| | **R2D2** | 7.27 | 4.70 | 9.94 | 16.94 |
| miniImageNet | **Baseline** | 19.14 | 13.01 | 17.90 | 21.65 |
| | **Baseline++** | 18.18 | 11.51 | 42.82 | 40.10 |
| | **ProtoNet** | 16.55 | 8.95 | 16.14 | 26.46 |
| | **MAML** | 26.14 | 13.61 | 42.38 | 43.48 |
| | **R2D2** | 16.83 | 8.96 | 16.36 | 23.06 |
| MetaAlbum | **Baseline** | 21.50 | 9.39 | 11.32 | 14.50 |
| | **Baseline++** | 20.29 | 9.32 | 12.22 | 14.91 |
| | **ProtoNet** | 15.78 | 9.06 | 9.77 | 10.58 |
| | **MAML** | 27.38 | 9.75 | 35.50 | 17.67 |
| | **R2D2** | 16.73 | 9.50 | 11.66 | 14.32 |

**miniImageNet**    This is a scaled down version of ImageNet consisting of 100 classes, each with 600 total examples and image scaled down to $84 \times 84$ pixels. We make use of the standard meta-train/validaiton/test split proposed by (Ravi & Larochelle, 2017).

**CIFAR-FS**    Uses the CIFAR-100 dataset (Krizhevsky, 2009), with 64, 16 and 20 classes for meta-training, meta-validation, and meta-testing, respectively.

**Meta-Album**    This is a recently proposed (Ullah et al., 2022) dataset that has a focus on providing data from a broad range of application domains, accomplished by pooling data from 30 different existing datasets. We use the Mini version of the dataset, which provides 10 different domains (three source datasets per domain) and 40 examples per class. Three splits are present, each containing one source dataset from each domain. Each split is used once each as meta-train, meta-validation, and meta-test, and the final evaluation results are obtained by computing the mean of these three runs.

**CLIP Few-Shot**    For experiments with CLIP-like few-shot learners, we also use three datasets (EuroSat, Flowers, and Food101) selected from the CLIP few-shot suite (Radford et al., 2021). In this case we follow the CLIP few-shot learning protocol, drawing from 1-16 shots from all categories in each dataset, and evaluating on the full test sets.

**Implementation Details**    We primarily study the classic FSL methods from the meta-learning community ProtoNet (Snell et al., 2017), MAML (Finn et al., 2017), R2D2 (Bertinetto et al., 2019), and Baseline(++) (Chen et al., 2019), using the implementations provided by the LibFewShot package.[1] We use the meta-training hyperparameters suggested in the documentation of this implementation, which were tuned using the meta-validation set of miniImageNet. We use the standard Conv4 architecture for miniImageNet and CIFAR-FS, ResNet18 for meta-album. From the vision & language community, we also study CLIP (Radford et al., 2021), using the VIT-B32 vision encoder architecture.

## 4.1 PERFORMANCE ESTIMATION

**Experimental Setup**    For each meta-test episode, all evaluators are given only the support set and required to produce an estimate of the generalisation error for models trained using each of the FSL algorithms we consider. We additionally construct a query set that is several times larger than the support set, and is used as an oracle estimate for the performance of each model. For each evaluator we construct a sample of performance estimates, where each data point corresponds to a meta-test episode. We report the mean of the accuracy across episodes, and also the mean absolute difference in accuracy between each estimator and the oracle accuracy.

**Results**    For the 5-way 5-shot setting, Table 1 compares FSL accuracy as observed by the oracle with the four standard estimators considered. We can see that the estimators tend to under-estimate the actual accuracy of the oracle, to varying degrees. This is due to the performance estimators

---

[1] https://github.com/rl-vig/libfewshot

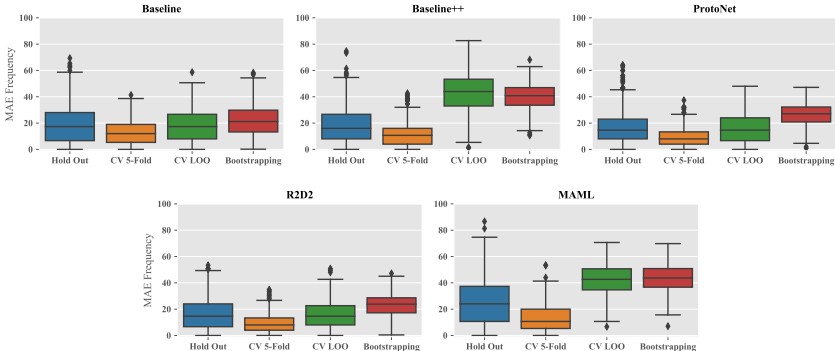

Figure 2: Box plots showing the distribution of absolute differences between the estimated accuracy and oracle accuracy on the meta-test episodes of miniImagenet. Distributions should be ideally concentrated as close to zero as possible, but we can see that a substantial proportion of the mass is far away from zero. This indicates that many of the performance estimates are unreliable.

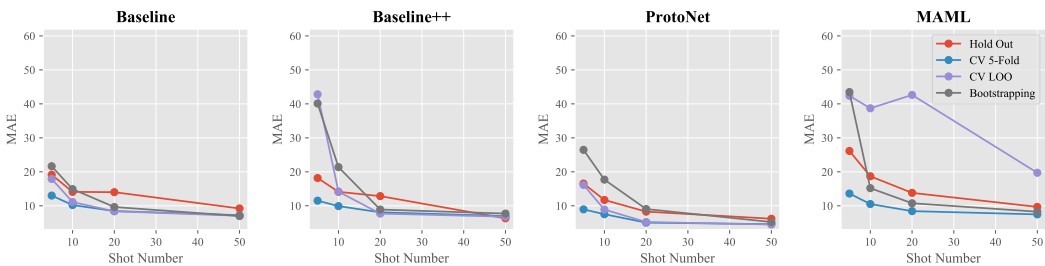

Figure 3: Dependence of estimator-oracle error on shot number. Estimator error is substantial in the few-shot regime.

having to sacrifice some of the training data in order to estimate performance, meaning the model fit is poorer than a model that is trained on all available data. This issue also exists in the many-shot setting, but the effect is usually negligible in that case because of the diminishing returns of training on more data.

Table 2 presents the mean absolute difference between each estimator and the oracle. We can see that 5-fold CV tends to have the lowest difference, i.e., it provides the best estimate of the oracle accuracy. We note, however, that it is still likely to be too high to be practically useful in many applications. To investigate this on a more fine-grained level, we visualise the distribution of absolute differences in Figure 2. From this figure we can see that the distribution of errors is quite dispersed for all combinations of learning algorithm and evaluator: *for all potential choices there is at least a 50% chance the validation accuracy would be wrong by more than 10%, in absolute terms*. This means that there is no pipeline that one can use to reliably train and validate models to ensure they are safe to deploy in situations where one must take into account the risks involved in deploying their models.

We next analyse how the error between oracle and estimator varies with number of shots. From the results in Figure 3, we can see that the error of the estimator decreases with shot number as expected, but tends to be substantial in the $\leq 20$ shot regime typically considered by FSL. Together, Table 2, Figure 2, and Figure 3 illustrate our point that there is no good existing solution to task-level FSL performance estimation.

### 4.1.1 FSL WITH VISION & LANGUAGE FOUNDATION MODELS

In Figure 5 we plot the number of shots versus the MAE and accuracy, measured on CLIP FSL problems. The issues seen in the meta-learning setting are less pronounced here—particularly in the Food101 and Flower102 datasets. This is because the FSL protocol in the CLIP setting uses all

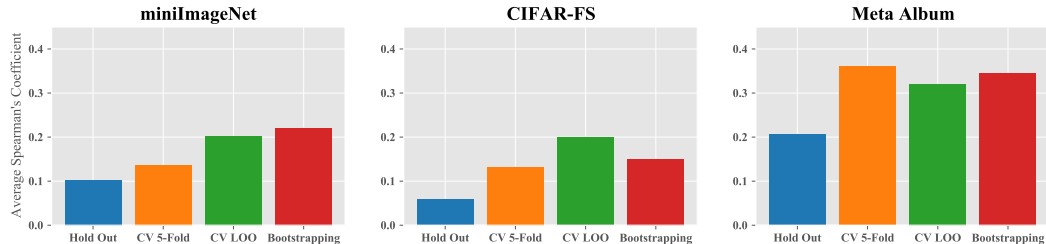

Figure 4: Mean Spearman correlation between the rankings produced by the oracle and the different performance estimators, computed across all the meta-test episodes in each dataset. A correlation coefficient of 1 indicates the same rankings, -1 indicates the opposite rankings, and 0 indicates that the rankings are unrelated.

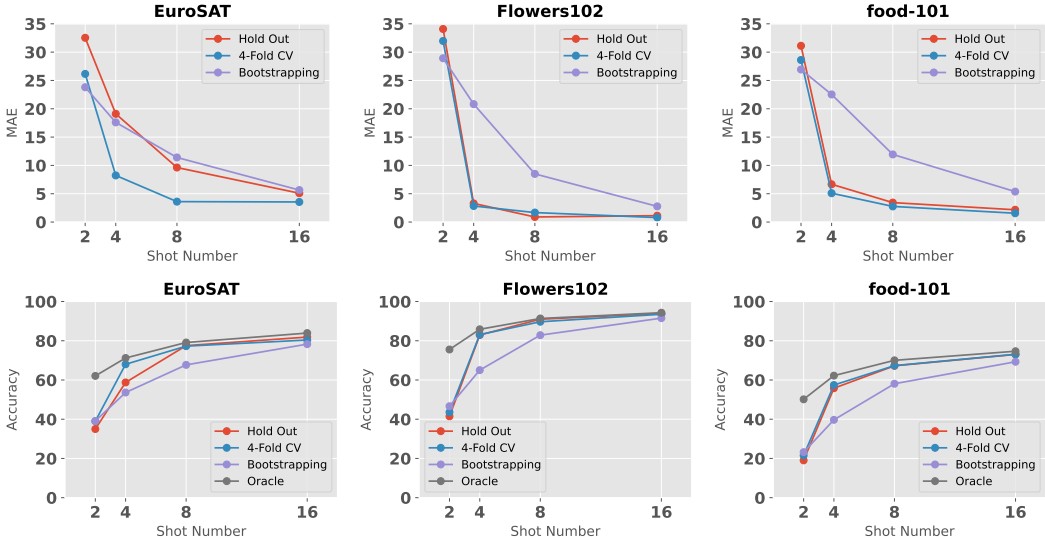

Figure 5: Few-Shot CLIP MAE (top) and accuracy (bottom) of different performance estimators averaged over 10 runs.

classes in these datasets, which increases the support set size compared to the meta-learning setting. E.g., with $k = 4$ the support set of a FSL problem from Flowers102 will have $408$ instances, which is considerably larger than the support set sizes typically considered in few-shot learning.

## 4.2 MODEL SELECTION

An alternative to accurate performance estimates that is acceptable in some situations is to instead rank a set of prospective models and select the one that performs the best according to some performance estimator. In this setting, the estimator can exhibit some types of biases without having an impact on the final rankings. For example, the performance underestimation bias of the estimators may not have a significant impact on the final rankings, as long as all models are similarly affected by this bias. However, the ranking procedure is still sensitive to variance of the estimators.

To this end, we investigate how well the rankings produced by the different estimators are correlated (in the Spearman Rank correlation sense) with the rankings produced by the oracle estimator. Rankings are produced for each episode in the meta-test sets, and the mean Spearman rank correlation (w.r.t. the oracle) is computed for each performance estimator.

**Results** The results of this experiment are shown in Figure 4. It is evident from these plots that no evaluator produces rankings that are highly correlated with the oracle rankings, with a maxi-

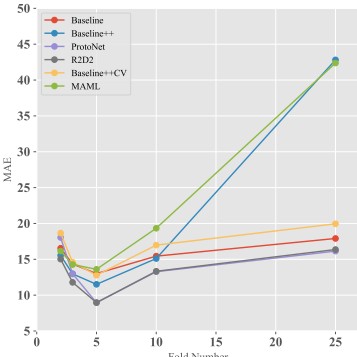 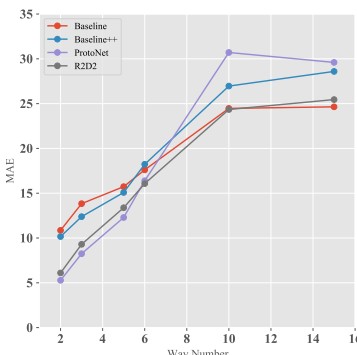

Figure 6: Further analysis of the estimation accuracy of cross validation performed on the miniImageNet dataset. Left: MAE between $k$-fold CV estimates and the oracle estimates, as a function of $k$. Right: MAE of LOO-CV, relative to the oracle, as the number of ways (and therefore class imbalance) is increased.

mum achieved correlation of around only 0.4 for bootstrapping on the meta-album benchmark, and resampling methods with a large number of iterations generally being the most reliable approach. However, these results indicate that we are unlikely to be able to do effective model selection using currently available evaluation procedures.

### 4.3 FURTHER ANALYSIS OF CROSS-VALIDATION

In the previous section showed 5-fold CV is the least bad existing estimator for task-level FSL performance estimation. This might seem surprising, as LOO-CV is generally considered preferred from a statistical bias and variance point of view (Witten et al., 2016). To analyse this in more detail, we plot the error as a function of folds in Figure 6 (left). The results of this show that increasing the number of folds—and therefore reducing the bias introduced by fitting each fold on a smaller training dataset—is counteracted by another phenomenon that causes LOO-CV to have low quality performance estimates. We find that 5-fold CV presents the ideal trade-off between these two effects.

We hypothesise that the negative effect that causes CV with a large number of folds to become less accurate is related to the class imbalance. Consider the standard balanced few-shot learning experimental evaluation setting. In the case of LOO-CV, the consequence of holding out one example for evaluation will mean that the training data contains one class with fewer examples than the other classes. Moreover, the test example will come from this minority class, which will result in LOO-CV evaluating the model in the pathological case where all test examples belong to the minority class encountered during training.

To investigate this hypothesis, we vary the number of ways while keeping the size of the support set fixed. As the number of classes is reduced, the number of training examples per class is increased, and class imbalance is less prevalent. Figure 6 (right) shows the result. Using the number of ways as a proxy for class imbalance in the LOO-CV setting, we see there is an association between class imbalance and MAE of LOO-CV performance estimation.

### 4.4 HOW CAN WE IMPROVE FSL IN PRACTICE?

We further investigate how existing model selection methods applied at the task-level can be used to improve performance. Experiments are conducting on both types of model selection seen in machine learning: hyperparameter optimisation, and algorithm selection.

**Hyperparameter Optimisation** As a compromise between computational cost and model rank correlation, we propose a new baseline for FSL that uses 5-fold cross validation to tune the ridge regularisation parameter of a Baseline (i.e., logistic regression) model (Chen et al., 2019). The results for this experiment are shown in Table 3. From these results we can see that using 5-fold CV can provide a noticeable improvement over the aggregated accuracy of the standard Baseline.

Table 3: Aggregated Accuracy of the different baseline models. BaselineCV indicates that the ridge regularisation hyperparameter is tuned on a task-level basis using 5-fold cross validation.

| Model | CIFAR-FS | miniImageNet | Meta-Album |
|---|---|---|---|
| Baseline | $71.17 \pm 0.727$ | $59.36 \pm 0.646$ | $59.36 \pm 1.688$ |
| BaselineCV | $72.89 \pm 0.737$ | $62.11 \pm 0.698$ | $58.46 \pm 1.745$ |
| MAML | $69.51 \pm 0.762$ | $59.76 \pm 0.717$ | $47.10 \pm 1.736$ |
| MAMLCV | $40.24 \pm 0.840$ | $55.06 \pm 0.849$ | $52.77 \pm 1.308$ |

Table 4: Aggregated Accuracy of Task-Level Model Selection using each of the performance estimators.

| Model | CIFAR-FS | miniImageNet | Meta-Album |
|---|---|---|---|
| Oracle | $80.11 \pm 0.495$ | $71.40 \pm 0.464$ | $63.53 \pm 0.932$ |
| Hold-Out | $71.65 \pm 0.749$ | $62.79 \pm 0.710$ | $56.30 \pm 1.048$ |
| 5-Fold CV | $72.38 \pm 0.726$ | $63.73 \pm 0.734$ | $58.58 \pm 1.005$ |
| LOO-CV | $73.29 \pm 0.738$ | $64.34 \pm 0.717$ | $58.53 \pm 1.014$ |
| Bootstrapping | $73.44 \pm 0.724$ | $64.05 \pm 0.737$ | $58.62 \pm 1.011$ |

**Algorithm Selection** By using each estimator to rank the performance of each FSL algorithm on a per-episode basis, we can try to select which algorithm should be used in each episode to maximise performance. E.g., MAML might perform best in one episode and Prototypical Networks in another. Table 4 shows aggregated accuracy results when using each estimator to do this type of model selection. These results demonstrate that LOO-CV and Bootstrapping are the best approaches to use for model selection, which is consistent with the rank correlation results from Figure 4.

## 5 DISCUSSION

This paper investigates the effectiveness of methods available for evaluating and selecting models on a task-level basis in few-shot learning. The experiments conducted in Section 4 provide several concrete takeaway points that can be used to inform best practices and show the limitations of the current state-of-the-art. Revisiting the questions asked in Section 1, we provide three main insights.

**Performance Estimation** It is demonstrated that existing evaluation approaches are not able to provide accurate estimates of the performance of models in the few-shot learning setting. For most combinations of learning algorithm and evaluation method, the performance estimates obtained for individual episodes are more than 10% away from the true accuracy, in absolute terms, for the majority of the episodes we generate. Even the best combination is only below this threshold approximately 50% of the time. Such unreliable estimates mean few-shot learning can not be validated for use in practical applications due to the unacceptable risk of performance being substantially different than estimated by any validation procedure. However, this risk can be mitigated by noting that some performance estimators such as LOO-CV and Bootstrapping consistently underestimate the accuracy. Such pessimistic estimators may be acceptable in some scenarios.

**Model Selection** We investigate how well these inaccurate performance estimates can be used to rank models, rather than provide precise estimates of performance. Such rankings would at least allow practitioners to select the best possible model, even if the exact performance is not known. It would also enable a better degree of hyperparameter optimisation than is currently employed in many FSL settings. Our findings in this area show that there is only a mild correlation between the best model ranking approaches and the oracle rankings provided by larger held-out datasets that are not typically available in the few-shot setting. Despite the only mild correlation, we do achieve a small degree of success in using these rankings for hyperparameter tuning.

**What makes a good performance estimator?** We argue that the stability of the underlying learning algorithm is an important factor when estimating performance. Many of the existing estimators rely on constructing a model using different subsets of the available training data and then aggregating several different performance estimates. Algorithmic stability has been shown to reduce the dependence on data when fitting models (Bousquet & Elisseeff, 2002), and more stable algorithms have also been shown to provide more reliable performance estimates when used in conjunction with cross validation (Aghbalou et al., 2022). This line of reasoning is congruent with our empirical results, where we see that the most stable algorithm (Prototypical Networks) consistently has low MAE (relative to other approaches) when coupled with CV estimator.

**Recommendations for Future Work** Just as few-shot learning requires specialised algorithms that take advantage of learned prior knowledge, we propose that future work should design specialised evaluation procedures. This could take the form of Bayesian estimation procedures that are informed by performance on the meta-training and meta-validation episodes, or they could leverage other side-information to reduce variance in estimates.

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
