# OpenReview forum: "Evaluating the Evaluators: Are Current Few-Shot Learning Benchmarks Fit for Purpose?"
_ICLR.cc/2024/Conference — Submitted to ICLR 2024_

### Official Review · Reviewer_Unr2 · 2023-10-31

**Soundness:** 3 good
**Presentation:** 3 good
**Contribution:** 2 fair
**Rating:** 3
**Confidence:** 3

**Summary:**

The paper aims to investigate 3 main research questions about the effect of evaluation approach on 1) predicting task-level performance, 2) validity of the ranking of various FSL methods and 3) model selection and FSL performance. Among other things, they concluded that performance is substantially different than the performance estimated by the validation procedures used in the paper. Among them, 5-fold CV is better than the others. For model selection LOO-CV is the best approach.

**Strengths:**

The paper is about a topic that hasn’t gotten much attention in the past.

**Weaknesses:**

Different methods use different backbone networks. Backbone network has a confounding effect that makes comparison between methods difficult.
Oracle estimator is not properly described.
My main concern is about the significance of the results and their usefulness and impact in real-world. For example, I am not sure to what extent it can be justified to do LOO-CV for model selection and then 5-old CV for performance estimation.

**Questions:**

See above

---

> ### Author Response · Authors · 2023-11-23
>
> We thank the reviewer for taking the time to provide feedback for our paper. We address the questions and concerns raised in the review below.
>
> **Q1: Different methods use different backbones?**
>
> **A1:** We did use the same backbone architecture for each method, so our results are not impacted by the potential confounder pointed out by the reviewer.
>
> **Q2: What is the oracle estimator?**
>
> **A2:** We described the oracle estimate as the performance measured on a query set that is several times larger than the support set. This is how all previous FSL literature has evaluated FSL methods, so we did not see the need to go into additional detail.
>
> **Q3: Significance of results?**
>
> **A3:** For many real-world applications one must know the performance of a model before deploying it; our main contribution is to show that it is not possible to reliably evaluate models in the FSL setting. We specifically say that no techniques are good for either model selection or performance estimation, thus motivating the development of new FSL evaluation methods by the research community.

---

### Official Review · Reviewer_CrkK · 2023-11-03

**Soundness:** 3 good
**Presentation:** 3 good
**Contribution:** 3 good
**Rating:** 8
**Confidence:** 4

**Summary:**

This paper tackles the problem of validating the performance of few-shot learning (FSL) algorithms, on a given task (or episode), given only the small number of example in that task's train set (support set). This is a different endeavour than the aggregated performance usually reported to compare few-shot learning algorithms in general: it is usually estimated over many episodes, on much larger test sets (query sets) to reduce variance.

Experiments compare different estimator of the generalization accuracy that only depend on the support set (hold-out, 5-fold and leave-one-out cross-validation, bootstrapping), against the accuracy as measured on the larger query set. They observe a large difference, which show none of these methods are reliable enough to be used as estimates of the generalization performance on a given task. Moreover, there is not much correlation between _rankings_ of models (or hyper-parameter values) on validation and test. However, these validation methods are usually _underestimating_ the test accuracy, so they may be useful as a lower bound (as the train error is usually an upper bound).

**Strengths:**

Originality
--------------
The question of a validation procedure is often neglected in few-shot learning, and proponents of a new algorithm often only focus on the aggregate test performance. This is an issue when trying to apply these algorithms to new, specific small datasets, and trying to determine the best learner (and hyperparameters) for them.

Estimators of generalization errors considered are not new (this was much more popular in machine learning when all datasets were much smaller than today), but systematically investigating them in the context of FSL is new, as are the observations on leave-one-out in that context.

Quality
----------
The research questions are clear, and explore well different aspects of the main problem (task-level validation for FSL), which is well motivated.
The investigation is well done, experiments align with the research questions.
The span of experiments, across datasets, models, and estimators make sense, and the results support the conclusions.

Clarity
---------
The paper is quite clear and reads well. Often questions or remarks coming to my mind when reading a sentence were satisfactorily addressed in the next one or a paragraph later.
Visualization and presentation of results are mostly clear.

Significance
-----------------
Although some of the conclusions may read like negative results, which are usually harder to "sell" as significant, I think this investigation is reveals really important points for any application of FSL to real world small-scale datasets. The issues exposed would affect anyone needing to validate the performance of a few-shot learner without access to a large labeled query set.
Episode-level hyper-parameter selection is also a pretty open problem, and it's good to have it explored.

**Weaknesses:**

1. One thing that could be explored further is the reliance of some learners on the assumption of a balanced support set. MAML seems to be particularly sensitive, as shown in Fig. 3 for CV LOO. This makes me wonder if maybe the "Oracle" accuracy would only be valid or accurate for balanced, N-ways k-shot support sets, and CV LOO or bootstrapping estimators might actually be closer to the performance of these few-shot learners on unbalanced support sets.
2. If class balance is an assumption that can be made (based on the composition of a given task), then maybe estimators could be adapted to have splits respect that constraint. Cross-validation could have leave-one-shot out (the support set would be balanced, k-1 shot and the valid set would have one example of each class), sampling for bootstrapping could also be aware of classes. If it works, this might be a practical contribution.
3. The class balance assumption, used in all datasets considered (as far as I can tell), may be too restrictive when developing procedures for real-world datasets.

**Questions:**

1. Could you also report training accuracy, for the different estimators (incl. oracle)? Or is it almost always 100% and there's no signal there?
2. Is it the case that all the estimators actually underestimate the test accuracy (up to statistical noise)?
3. If we can assume that tasks of interest have balanced training sets, can class-aware splits provide a better estimation? (see point 2 in "Weaknesses" above?
4. If class balance cannot be assumed in general (as I'd think would be the case in real world applications), maybe it would be worth doing experiments on episodes with un-balanced support sets, either from other benchmarks, or altering the composition of episodes from usually-balanced ones.

Update after reply
------------------------
The author reply did not add much information, confirming observations and repeating points made in the paper, but did not provide additional insight, theoretical arguments, or observations. Therefore I'm maintaining my score.

---

> ### Author Response · Authors · 2023-11-23
>
> We thank the reviewer for taking the time to provide feedback for our paper. We address the questions and concerns raised in the review below.
>
> **Q1: Training accuracy?**
>
> **A1:** This is a good suggestion. We plan to add support set accuracy as an evaluator in the final copy of the paper.
>
> **Q2: Do estimators underestimate the test accuracy?**
>
> **A2:** From Figure 1 we can see that methods are more likely to underestimate the accuracy than to overestimate it.
>
> **Q3: Stratified cross-validation?**
>
> **A3:** We agree it is likely that some of the performance degradation in the balanced setting comes from class imbalance introduced by the splitting procedures. Section 4.3 (Figure 6 right, in particular) provides some supporting evidence for this.
>
> **Q4: Can class balance be assumed in general?**
>
> **A4:** We agree that class imbalance is a common feature of real-world problems, and experiments in this area would be interesting. We investigated the balanced setting because that is what FSL literature typically focuses on.

---

### Official Review · Reviewer_ZAXf · 2023-11-08

**Soundness:** 3 good
**Presentation:** 2 fair
**Contribution:** 1 poor
**Rating:** 3
**Confidence:** 2

**Summary:**

This paper explores the reliability of evaluation methods for few-shot learning (FSL) models. The authors investigate how well current benchmarks and validation strategies predict the performance of FSL models on individual tasks. They find that cross-validation with a low number of folds is best for direct performance estimation, while bootstrapping or cross-validation with a large number of folds is more suitable for model selection purposes. However, they conclude that with current methods, it is not possible to get a reliable picture of how effectively methods perform on individual tasks.

**Strengths:**

+ The paper addresses an underexplored and realistic aspect of few-shot learning—task-level validation.
+ This paper is easy to read.
+ It provides an extensive experimental analysis across different datasets and evaluators, offering a comprehensive view of the current state of FSL benchmarks.
+ The paper identifies the best performing evaluation method, which is useful for future research and practical applications.

**Weaknesses:**

Firstly, it is hard for me to estimate the novelty and contributions of this paper. The paper is more like an analysis paper but the conclusions are not clear. From the end of the introduction, all three conclusions are weak. Moreover,
- I don't fully understand the part related to the first question: all three evaluations (hold-out, cross-validation, and Bootstrapping) are widely used for few-shot evaluations. The only difference is using the support set only. But if we treat the support set as the whole set, there are no obvious differences.
- Secondly, I cannot agree with the hypnosis in Figure 1, since the oracle set is larger than the estimate set and, there are a lot of variances for each estimate set, it is not surprising that the performance on the oracle set is more stable.

A few other minor weaknesses include:
- Even the best evaluation strategy identified is not entirely reliable, indicating that current evaluation benchmarks may be inadequate for individual task performance prediction.
- Regarding few-shot learning, besides the standard few-shot evaluation benchmarks, more few-shot evaluations are flexible -- in a lot of papers they also show the performance under a few-shot setting with different synthetic or real-world benchmarks, even for open-vocabulary images/videos. They are also not limited to the standard query-support evaluation. How can the conclusion drawn from this paper inspire those types of work?

---
Thanks to the authors for providing the response. However, I don't think the response addressed my concerns.

For Q1, yes I agree the Aggregated Evaluation is what people mostly use for the ``standard'' few-shot setting but the other settings are also used for works of literature regarding few-shot learning. As the positioning of this paper is also NOT studying the standard few-shot problem, I don't think ignoring the evaluation methods used by other works of literature is fine.

For Q2 and Q3, the query set is used for evaluation but only the support set is used as labels seeing by the model under the few-shot setting. Therefore, I don't think a large query set is a big problem -- in real practice, the few-shot model should be able to learn from a few samples but can work with a large number of test samples. Moreover, if merging the query set with the support set, how should we evaluate the model? For the standard academic benchmarks, there is no real concept of a few-shot problem as we can always label more data for the few-shot categories. Providing enough labels and splitting them into query and support sets should be more considered for evaluation but not training.

Based on this understanding, I keep my original rating.

**Questions:**

A few minor questions:
- (Section 3, AE)"We observe that in a true FSL setting, this is an unrealistic assumption." --> Why this is not a realistic setting? For few-shot in practice, we want to tune a model with a few samples and that model can handle a lot of samples, which means the query set is much larger.
- (Section 3, TLE) "Moreover, in realistic situations there is no labeled query set for evaluating a model, so both the model fitting and model evaluation must be done with the support set." --> Again, query and support set split is some data processing. If we treat the support set as the whole labeled set, the held-out strategy is essentially the same as the query-support split, right?
- (Section 4.1) What are the evaluators? They should be clearly defined. Moreover, there are a lot of similar terminologies used in this paper, e.g. estimators, and oracle accuracy, which sound not general enough for an audience out of FSL.

---

> ### Author Response · Authors · 2023-11-23
>
> We thank the reviewer for taking the time to provide feedback for our paper. We address the questions and concerns raised in the review below.
>
> **Q1: All evaluation methods are widely used in FSL?**
>
> **A1:** They are not. The standard way that FSL methods are evaluated in the literature is via Aggregated Evaluation, where performance measurements are aggregated over a large number of meta-test episodes. In contrast, the evaluation protocol we investigate is task-level validation. Section 3 of our paper described this distinction in detail.
>
> **Q2: Not surprising that the query set is more stable?**
>
> **A2:** Large query sets are not available in real-world applications. The purpose of our paper, as discussed in the introduction, is to demonstrate that using only the support set for model training and validation results in performance estimates that are unstable to the point of being unusable. Figure 1 shows this.
>
> **Q3: Why is the existence of a large query set unrealistic in a true FSL scenario?**
>
> **A3:** If a larger labelled query set was available then these points could be combined with the support set to form a dataset that is too big to be considered “few-shot”.
>
> **Q4: What are the evaluators?**
>
> **A4:** They are described in Section 3 under the Task-Level Evaluation heading.

---

### Official Review · Reviewer_KXUd · 2023-11-08

**Soundness:** 3 good
**Presentation:** 2 fair
**Contribution:** 3 good
**Rating:** 6
**Confidence:** 2

**Summary:**

This paper aims at rethinking how we could better evaluate the performance of few-shot learning methods and how we could select models in few-shot settings. The authors measure the estimated accuracy from different estimators to show that they all have non-negligible gaps from the oracle estimator. Also, the authors investigate different ranking strategies for model selection in few-shot learning. Finally, the authors provide insights that existing evaluation approaches are all not competent enough. Future works may need to design specialized evaluation procedures for evaluating few-shot learning performance.

**Strengths:**

++ This paper provides some interesting analysis and viewpoint on the evaluation, model selection for few-shot learning tasks. It could bring researchers to take a step back and reconsider the essential parts in few-shot learning, like what is a more proper way to evaluate models, and how to select models that can be better used in real-world cases.

++ The experimental results are comprehensive, involving various meta-learning datasets and meta-learning algorithms, which enhances the soundness of the conclusions from this paper.

**Weaknesses:**

-- I am uncertain about how useful in real application would the conclusion from this paper be. After all the experimental investigation and analysis, the suggestion given by the authors is that every few-shot setting should design specialized evaluation procedure. This would be laborious and complicated for future works which makes this suggestion infeasible. Also, there is no example given by the authors about how to design the evaluation procedure based on certain specific tasks.

-- I assume "CV" is short for "cross-validation" and "CV LOO" means "cross-validation leave-one-out", right? However, there is no explanation in the paper about what they mean, which causes confusion.

**Questions:**

-- From the results in Tab. 1, it seems that all estimators generally have under-estimated accuracy compared with the oracle method, so all of them should be pessimistic. However, in the conclusion section, it seems that only some of the estimators like LOO-CV and Bootstrapping are pessimistic estimators. I am wondering whether there exist an inconsistency or I have misunderstood the results in Tab. 1 and the conclusion on "Performance Estimation".

---

> ### Author Response · Authors · 2023-11-23
>
> We thank the reviewer for taking the time to provide feedback for our paper. We address the questions and concerns raised in the review below.
>
> **Q1: How useful for real applications?**
>
> **A1:** In many real applications it is important to know the performance of a model before deploying it. Our work demonstrates that existing evaluation procedures are not suitable for the standard FSL setting, so new methods must be developed. Whether this is laborious or not does not change the situation about the need for such evaluation methods. Moreover, we suggest developing evaluation methods based on Bayesian inference, as this paradigm is already widely used in the FSL community for developing new learning algorithms for various settings.
>
> **Q2: What are CV and CV-LOO?**
>
> **A2:** Yes, CV and CV-LOO are short for cross-validation and cross-validation leave-one-out, respectively. We have updated the paper to make this clear.
>
> **Q3: Are only CV-LOO and Bootstrapping pessimistic?**
>
> **A3:** All evaluators are pessimistic in expectation (i.e., the estimators are biased), but LOO-CV and Bootstrapping have lower variance as well. This means the risk of overestimating performance on an individual episode is lower than it is for, e.g., hold-out. Figure 1 provides a good visualisation of this.

---

### Meta-Review · Area_Chair_2noQ · 2023-12-12

**Metareview:**

Summary: This paper studies an under-explored aspect of few-shot learning (FSL) – how to reliably evaluate the performance of FSL models and perform model ranking and selection for a specific task/episode, given a few examples in the support set of that task. This contrasts with the common practice of aggregating performance over numerous tasks/episodes, termed the oracle evaluator. Different estimation strategies are investigated, including hold-out, 5-fold cross-validation, leave-one-out cross-validation, and bootstrapping. Empirical results indicate that these estimators yield poor classification accuracy estimates, exhibiting noticeable gaps from the oracle evaluator's accuracy. Furthermore, the rankings generated by these estimators show no correlation with those of the oracle evaluator. Among these non-reliable estimators, 5-fold cross-validation emerges as the most effective for performance estimation, while leave-one-out cross-validation and bootstrapping carry a lower risk of overestimating performance. Overall, the study underscores the unreliability of existing FSL evaluation procedures in providing task-level performance estimates, highlighting the need for significant future work in this direction.

Strengths and Weaknesses: The reviewers generally recognize that this paper addresses a crucial yet under-explored problem in FSL – specifically, the reliability of task-level performance evaluation and model selection. They also appreciate the thorough empirical analysis conducted in this paper across multiple datasets and estimators.

However, they express reservations about the significance of the findings and their practical usefulness. Specifically, all the estimators investigated in the paper exhibit poor performance. While 5-fold cross-validation outperforms other estimators for accuracy estimation, and leave-one-out cross-validation outperforms other estimators for model selection, their performance, significantly lagging behind the oracle evaluator, raises doubts about their utility in real-world scenarios. Additionally, the reviewers are concerned that the study in this paper is confined to episode-based FSL paradigms and may not be generalizable to other FSL settings in a broader context. Moreover, there are questions about the underlying assumptions of the findings, such as class balance in support sets.


The authors addressed some of these points during the discussion phase. However, the reviewers remained unconvinced and were not championing the paper. While the study's findings are interesting and expose the limitations of existing FSL evaluation mechanisms, there is a lack of an effective solution proposed. Towards the end of the paper, the authors suggest designing specialized evaluation procedures, potentially in the form of Bayesian estimation procedures, which is an intriguing avenue but could benefit from substantial expansion. It would also be interesting to conduct additional investigations under class imbalance scenarios.

Therefore, the paper is not ready for this ICLR. I encourage the authors to continue this line of work for future submission.

**Justification For Why Not Higher Score:**

The current recommendation is grounded in the identified weaknesses, notably the lack of convincing significance and practical usefulness of the findings, as well as the absence of an in-depth exploration of potential solutions.

**Justification For Why Not Lower Score:**

N/A

---

### Decision · Program_Chairs · 2024-01-16

Reject